# Low-Cost Sensors for Urban Noise Monitoring Networks—A Literature Review

**DOI:** 10.3390/s20082256

**Published:** 2020-04-16

**Authors:** Judicaël Picaut, Arnaud Can, Nicolas Fortin, Jeremy Ardouin, Mathieu Lagrange

**Affiliations:** 1UMRAE, Univ Gustave Eiffel, IFSTTAR, CEREMA, F-44344 Bouguenais, France; Arnaud.Can@univ-eiffel.fr (A.C.); Nicolas.Fortin@univ-eiffel.fr (N.F.); 2Wi6Labs, F-35510 Cesson-Sévigné, France; jeremy.ardouin@wi6labs.com; 3LS2N, UMR CNRS 6004, Ecole Centrale de Nantes, F-44321 Nantes, France; mathieu.lagrange@cnrs.fr

**Keywords:** noise, low-cost sensors, networks

## Abstract

Noise pollution reduction in the environment is a major challenge from a societal and health point of view. To implement strategies to improve sound environments, experts need information on existing noise. The first source of information is based on the elaboration of noise maps using software, but with limitations on the realism of the maps obtained, due to numerous calculation assumptions. The second is based on the use of measured data, in particular through professional measurement observatories, but in limited numbers for practical and financial reasons. More recently, numerous technical developments, such as the miniaturization of electronic components, the accessibility of low-cost computing processors and the improved performance of electric batteries, have opened up new prospects for the deployment of low-cost sensor networks for the assessment of sound environments. Over the past fifteen years, the literature has presented numerous experiments in this field, ranging from proof of concept to operational implementation. The purpose of this article is firstly to review the literature, and secondly, to identify the expected technical characteristics of the sensors to address the problem of noise pollution assessment. Lastly, the article will also put forward the challenges that are needed to respond to a massive deployment of low-cost noise sensors.

## 1. Introduction

Noise pollution is a major environmental pollution whose impact on health is now widely recognized [1]. As a result, many countries have implemented policies and strategies, for many years, to reduce noise pollution and to preserve quiet areas. Moreover, these policies are increasingly introducing citizens into the actuation process, by giving them the opportunity to be informed about their exposure to noise but also to be active in the management of their noise environment. In Europe in particular, Directive 2002/49/EC [2] introduces many rules on the assessment and management of noise environments, including the production of strategic noise maps, which are the starting point for the implementation of action plans to reduce noise pollution, but also as a tool for communicating between the different stakeholders. While this directive applies to large European cities, their general application to any community is obviously permitted.

The production of realistic noise maps is therefore a major challenge to ensure the proper implementation of the European directive, as well as, the relevance of the proposed actions to control noise pollution. Currently, noise maps are obtained by using sound mapping software tools, based on standards for calculating acoustic emission and propagation in the environment. As soon as the input data that are necessary for the calculation are available, the undeniable advantage of this method is its ease of implementation, making it possible to produce noise maps over a very large area. Auditing action plans using such noise maps can only be relevant if the limits of the methodology considered to produce the maps is taken into account during the process. In particular, only transportation noise sources and industrial noise are taken into account, on the basis of noise emission data that are very largely simplified and averaged over a large time period. It can also be pointed out that the propagation models used in software are based on many approximations (for example, lack of consideration of diffusion through facades and fitting objects; limited consideration of urban micro-meteorological conditions and vegetation). In addition, such noise maps cannot account for temporal dynamics, which nevertheless plays a role in the way inhabitants perceive the quality of sound environments [3,4].

In situ measurements would therefore be an immediate solution to make these maps more realistic. Nevertheless, given the urban scale considered and the spatial variability of the sound environment, the number of measurement points to be considered would be very large to model the relevant variability of the sound environment [5]. While noise observatories currently exist in many cities around the world, with networks of up to 150 measurement points [6], their use is mostly intended to provide objective and enforceable data on sound environments in certain strategic locations in order to better understand why inhabitants of this area are concerned with the quality of their sound environment. In view of the cost of a professional measurement point using Class-1 devices [7], enlarging the number of sensors of this type of measurement networks would be very costly. As a consequence, it is not realistic to aim at producing strategic noise maps on the technical basis of such observatories.

Many major technical developments have emerged in the last decade, making it possible to develop capturing devices integrating transducers of different kinds, embedded processing systems and wired/wireless communication systems, while optimizing power consumption and reducing their size. At the same time, this technological development has been accompanied by a significant reduction in the costs of electronic components and products, making these capture devices more affordable. Consequently, environmental monitoring tools based on Internet of things (IoT) have emerged in many related areas: atmospheric pollution, agriculture, transportation, smart cities [8,9,10]. Application of IoT to the monitoring of the sound environment has also a great potential. However, it should be noted that the development of low-cost sensors for acoustic measurement and the deployment of an ad-hoc sensor network can be complex to implement. Indeed, the high spatial and temporal variability of sound environments requires a high density of sensors, and advanced processing capabilities within the sensors [11].

Thus, the use of low-cost Sensor Networks (SN) can be a solution to the current limits of the noise observatories, mentioned above, by making it possible to reach a density of measurement points in a territory that is capable of providing a very rich acoustic information. The use of such a noise SN also opens up many additional interesting opportunities, such as to assess and manage road traffic noise [12,13,14], to enhance the traditional strategic noise mapping process [15,16,17,18,19], to produce dynamics noise maps [20] or to capture the sound sources of interest or acoustic events within the signal [21,22,23,24,25,26,27,28,29]. One can mention also that low-cost SN may be used in other fields in acoustics, such as sound source localisation [30] and biodiversity monitoring [31,32,33].

The relevance of such a system relies on many elements, among which, the acoustic measurement quality, the resilience of the sensors, the implementation of a communicating and smart sensor network, the deployment of a compatible data infrastructure able to manage the considerable amount of obtained information, the maintenance of the devices and infrastructures, the development of powerful algorithms to manipulate data and many others. Each of these topics are the subject of a considerable number of articles in the literature in the field of sensor networks for environmental monitoring. Being a more recent subject of the application of sensor networks, the literature specific to the sound environment subject is currently focused on the development of sensors themselves, but it is clear that the other subjects will see a growing of interest in the future.

The purpose of this article is thus to focus on the main element of such data gathering systems, namely the sensor. Solutions for the implementation of a complete information technology (IT) infrastructure for sensor, network and data management is not developed in this article, as this topic would in itself merit its own literature review in order to address the important issues that are associated (data encryption, privacy, security, scalability, management of a large number of sensors, interfacing, database management, storage, data visualization, etc.). Nevertheless, the reader can already refer to recent publications showing examples of full noise SN deployment [34,35].

For the past fifteen years, many researchers have proposed, evaluated, and in the best cases, deployed technical solutions for different noise applications. It therefore seems interesting to make a detailed point of current research on this subject, and in particular to highlight the essential characteristics that must be considered for this new generation of acoustic sensors in order to respond to the identified issues and the future ones. Compared to recent articles that already present a review of the existing system [35,36,37,38,39], the present contribution focuses on a detailed analysis of the technical solutions developed in the literature, highlighting their strengths and weaknesses, and showing how the rapid evolution of technologies can now fully meet the requirements for a successful deployment of modern noise SN.

The paper is organized as follows. Section 2 presents a review of the literature concerning the development of low-cost sensors for the deployment of noise monitoring networks. After a brief description of the SN terminology, the main technical solutions are detailed and synthesized. This synthesis then allows to identify the characteristics and performances that are expected for low cost sensors, in order to meet current and future noise monitoring challenges (Section 3). A conclusion then closes this article.

## 2. Literature Review

### 2.1. Sensor Networks: Definitions

A SN structure can be abstracted into a composition of a set of *nodes* and a *sink* node (more simply the *sink*, also called *gateway*. A node collects information, performs pre-processing and transmits the produced data to the sink. The sink collects all the data from the nodes and transmits them to servers for storage and further processing. On the basis of this simple architecture, many variants can be designed, depending on the needs and targeted applications, each variant giving rise to specific technical constraints. Among the most common variants are the following [40] (Figure 1):The network can be made up of several sinks. In this case, a group of identified nodes transmit the produced data to a specific sink. All sinks then transmit data to the servers. Another possible option is to consider that a node can choose the sink according to particular constraints, such as availability, proximity, sink load...The transmission of data from one node to the sink can be relayed using one or more nodes. The node then acts simultaneously as a sensor and a *relay*. This defines a *multi-hop* sensor network, as opposed to *single-hop* sensor network. The management of data transmission from the nodes to a given sink is then governed by relatively complex routing protocols that depend on the selected topology, such as point-to-point, star or mesh topologies.The nodes and the sinks may be mobile. The network is then defined *mobile* sensor network, as opposed to a *static* network. The term ’mobile’ must be considered in two ways: (1) continuously ’mobile’: a node moves continuously over time (like a sensor installed on a vehicle); (2) occasionally: a node is moved from one static position to another static position, for a long measurement time, in which case the network is always considered as a static network.Data transmission from a node to a sink can be performed in wired or wireless mode. In the latter case, the network is defined as a wireless acoustic sensor network (WASN). Nowadays, the wireless transmission mode is almost the main part of sensor networks for environmental monitoring. Data transmission from a sink to the server can also be carried out using one of these two transmission modes. Nodes and sinks may also simultaneously include several wireless transmission protocols, where some protocols get involved in the case of failure of the main protocol.The type of power supply of the nodes can also give rise to several variants: via a public or private power grid, by exchangeable battery, power supply by rechargeable battery from an external renewable energy source (solar, wind).Several families of nodes can also be considered, each with its own technical characteristics (measurement characteristics, processing power, power supply mode...). In this case, we are talking about a *heterogeneous* sensor network, to be opposed to a *homogeneous* network.

### 2.2. Low-Cost Noise Sensors: Literature Review

Since the mid-2000s, with the emergence of low-cost electronic components and computing processors, many researchers have been working on the development of low-cost noise SN, mainly for environmental acoustics applications. Three SN main families can be distinguished: (1) fixed sensor networks, (2) mobile participatory measurement networks (mainly with smartphones), (3) mobile sensor networks. In the context of this study, we mainly rely on low-cost sensors specifically developed for environmental acoustic measurements, allowing to set up a controlled measurement network. Participatory measurement with smartphones is an interesting approach, complementary to that of low-cost and professional static networks, but beyond the scope of this review. Nevertheless, it can be noted that the approach shares common technical aspects with static noise SN, such as data pre-processing, the choice of acoustic indicators, the implementation of a data management infrastructure [41] ... Lastly, the third family, concerning mobile sensors, although generating significant additional complexities in relation to data transmission and sensor autonomy, considers specifically designed noise sensors, and thus will also be included in this review.

The goal of the present section is to focus on references that give detailed information of the proposed low-cost noise sensors and additional information concerning their validation and tests. Some technical aspects are deliberately developed in order to highlight the diversity of the proposed solutions and to be able to subsequently identify the technologies, solutions and challenges that have strong potential for future research.

A first study about the use of sensor networks considered as an alternative to traditional “professional” networks for noise monitoring have been presented in [42,43,44]. Initially, the authors used a Tmote Sky platform (Moteiv), made of a microcontroller unit (MCU) (Texas Instruments MSP430 with a 16-bit RISC central processing unit (CPU)) equipped with a multimodality sensor board (EasySense SBT80) and an omnidirectional condenser microphone (ECM) (EM6050P-423) [43]. In a second step, they turned to a Tmote Invent platform made of a MCU (Texas Instruments MSP430) [42]. Sensors were connected to a sink via the IEEE 802.15.4 radio protocol. However, the authors have quickly highlighted the limitations of the microprocessors used (which limits the sample frequency to 8 kHz for the sound environment study), the rapid battery consumption at the relevant sampling frequencies and in simultaneous radio transmission, the dispersion and poor quality of the microphones used. They decided, in the first, instance to develop their own sound level meter [44], and later to use a Class-2 commercial sound level meter (Extech 407740) [36], both connected to the Tmote Sky platform via an analog input channel, mainly to overcome the problem of limited computational resources. Despite the technical limitations encountered, the authors showed that the use of wireless sensor SN was technically possible and could address the issue of environmental noise assessment in the future.

At about the same time, McDonald et al. [12] developed a noise sensor at a cost of about 130 GBP, based on a Triton XXS platform, using an Intel XScale PXA255 32-bit MCU. The acoustic measurement was performed through an ECM connected to the MCU via a 16-bit analog-to-digital converter (ADC) (sampling at 49 kHz). Equivalent sound levels over a given time period were calculated using a A-weighting digital filter. The sensors have been deployed on site, but unfortunately no information is given on the final acoustic characteristics of the sensors, their calibration and accuracy.

Although the first works were published from 2008, it seems that the origin of the studies conducted by Barham et al. [45,46,47] dates back to 2004. The work carried out seems to be particularly successful. The authors have very early highlighted the interest of using micro-electrical-mechanical systems (MEMS) microphones for environmental acoustic measurement, including their own design of a MEMS microphone with optimized performance for the environmental issue. In addition, numerous tests have been carried out to evaluate both the intrinsic acoustic performance of the MEMS used and the evolution of such solution under operating conditions (effect of atmospheric conditions, wind, cold, vibrations, etc.). The techniques developed were sufficiently advanced to deploy a hundred of sensors on several urban sites, and to start analyzing the collected data. Only little information is provided by the authors on the nature of the electronic components that where considered, excepted that they used a Floating-Point-Gate-Array (FPGA) system instead of a MCU, for an efficient use of the battery. Developments continued thereafter and allowed the implementation and deployment of nodes consisting of a specifically developed MEMS microphone package, connected to Raspberry Pi (R-Pi) 2 Model B with wired power supply, and using global system for mobile communications (GSM) connection [48].

One of the first works showing the design of a fully operational low-cost noise sensor network was detailed in [49,50,51]. The sensor was based on a CiNet platform, composed of a 8-bit MCU (Microchip ATmega128) with a 10-bit ADC running at a sample frequency of 33 kHz, using a IEEE 802.15.4 compliant chip (CC4420 Texas Instruments) and a high-quality omnidirectional ECM (MONACOR MCE-400). The authors implement a real-time A-weighting analog filter, as well as two pre-amplification channels (Maxim MAX4524) in order to overcome to the limited theoretical dynamic imposed by the ADC. Measurements were compared to Class-2 and Class-1 sound level meters, in indoor and outdoor environments, showing deviations of less than 2 dB. Beyond the technical aspect of the sensor developed, the work also focused on the development of a fully functional multi-hop large-scale network infrastructure, composed of nodes connected to a sink through relay nodes, all perfectly synchronized [52]. Later, the sensor network was successfully deployed for long-term measurements in a school yard, showing a good agreement by comparison with a Class-2 sound level meter [53].

In 2013, Tan and Jarvis [54] focused their works mainly on the energy harvesting using solar panel. The sensor node was based on an ultra-low power wireless sensor module TelosB [55] made of a 16-bit Texas Instruments MSP430 MCU, running on TinyOS, with the IEEE 802.15.4 protocol for the radio transmission and an electret microphone (Sparkfun BOB-09964). Various technical issues were highlighted, including the poor acoustic performances of the ECM in terms of signal-to-noise ratio (SNR) and dynamic, as well as the limitations of the MCU (memory size) and of the analog-to-digital converter (ADC). To overcome some of the limitations, the authors proposed latter to use a MEMS microphone (ADMP401) [56], offering better acoustic performances. The results mainly showed the need for sufficient energy storage capacity to ensure the proper functioning of the sensor over the duration of a measurement (depending of the kind of acoustic outputs that are required), as well as the need of power consumption management algorithms to perform autonomous measurements.

Mariscal-Ramirez et al. [57,58] presented in 2011 and 2015 the implementation of a fuzzy noise indicator and the development of frequency based algorithm for the calculation of noise indicators. The sensor node was based on a Sun Spot node with an electret microphone, using a 8-bit ADC and the IEEE 802.15.4 data transmission protocol. The node was tested on several signals, showing a deviation lower than 4% in term of sound levels, in comparison with a Class-2 sound level meter.

Segura-Garcia et al. [59] compared two low-cost acoustic measurement platforms, in terms of psycho-acoustic metrics instead of classical equivalent sound level. The first solution was based on the Tmote Invent platform as already used in [42]. The second was an R-Pi platform, composed of a Broadcom BCM2835 chip with an ARM1176JZF-S MCU and a Wi-Fi adaptater (IEEE 802.11 b/g/n). Acoustic Measurements were carried out using a universal serial bus (USB) sound card (Logilink UA0053) with an omnidirectional ECM. As already mentioned in [42], the Tmode Invent platform presents serious limitations for acoustic measurement, which definitively precludes it from any relevant operational solution. On the other hand, the data produced by the R-Pi shows a very good correlation with reference measurements made in the real-world, in terms of sound levels and psychoacoustic indices. This solution was used later to collect data in a city of Spain and to test a geo-statistical methodology for noise level prediction in urban areas [60]. The network was made of 39 nodes, some at fixed locations, others being moved to different locations. In order to collect and manage data, the authors successfully implement a specific network infrastructure, using a routing protocol (Babel/Quagga) for the wireless mesh network, a client–server suite (OwnCloud) for synchronizing data and a reliable and interoperable server (OpenCPU) for data analysis based on the statistical computing software R.

In 2016, Noriega-Linares and Navarro Ruiz [61] have proposed the design of an advanced acoustic sensor on the basis of R-Pi 2 Model B platform (Broadcom BCM2836 chip with ARM Cortex-A7 MCU) in order to compute in real-time several standard noise level indicators, such as instantaneous and equivalent sound levels, percentile levels, and 1/3 octave sound pressure levels. Unlike the solutions proposed so far, the authors have chosen a wired connection to the network, using a Power over Ethernet (POE) injector/splitter, which avoids potential interference between the hardware within the sensor and the radio transmission. Audio acquisition was carried out using a USB omnidirectional ECM (T-Bone GC 100 USB), including an internal ADC. According to the authors, this embedded solution offered better performances in terms of noise than using an external microphone with an USB sound card. Microphone frequency response equalisation, microphone calibration, 1/3 octave band filtering and real time implementation are mentioned by the authors, but are not detailed in their paper. The comparisons of measurements with a reference sound level meter seem to demonstrate a good performance of the sensor.

In a paper dating from 2016, Alsina-Pagès et al. [37] describes the design of a low-cost mobile noise sensors network. The corresponding article only describes the principle of a low-cost sensor and of the ad-hoc network, resulting from a detailed analysis of possible technical solutions. The proposed sensor is composed of a ARM 32-bit MCU on a NXP FRDM-KL25Z chip with a 12-bit ADC, using an omnidirectional electret microphone (CMA-4544PF-W) with a microphone amplifier (Maxim MAX9814 with automatic gain control). In order to optimize the data transmission from the mobile sensors to the data server, depending of the networks availability, the authors recommend to consider two data transmission protocols, a Wi-Fi connection (using the Wi-Fi ESP8266 module) and a GSM network (using the Adafruit FONA 808 cellular module, including a GPS). Since the design is very innovative, this research is worth mentioning. However, the implementation of this sensor in an operational system seems particularly difficult given the technical challenge of such a mobile network, such as ensuring a full and permanent connectivity between mobile nodes.

In 2017, Mydlarz et al. [62] have presented a low-cost noise sensor based on mini personal computer Tronsmart MK908ii with a Rockchip RK3188 quad core Cortex A9 CPU, a Wi-Fi connectivity and a wired connection to the public power supply. Acoustic signal acquisition was performed using an analog MEMS microphone Knowles SPU0410LR5H-QB with a eForCity USB audio interface on a specifically designed printed circuit board (PCB) and installed on a specific mount to limit potential harms due to weather. The set of acoustical tests show that the sensor generally meets the Class-2 criteria of the IEC 61672-1 standard [7]. This sensor was recently updated [35], replacing the processing unit by a R-Pi 2 Model B device in order to increase the computational performances and using a digital MEMS in order to reduce the radio-frequency (RF) interference. This sensor was successfully deployed across New-York City, producing a huge acoustic data collection in terms of sound level indicators and audio signal samples.

Although the sensors were designed for indoor use, the network presented in [38] in 2018 can be detailed here, since such device could be adapted for outdoor environments. In their paper, the authors use an analog MEMS microphone (SiSonic${}^{\mathrm{TM}}$ SPM0408LESH-TB) with an analog amplification gain of 20 dB, mounted on a MCU (ARM 32-bit Cortex STM32F050K6U6A, STMicroelectronics), while the sink node is carried out using a PC-based system on a R-PI with a ETRX357 ZigBee module. Although the authors pointed out the possible defects of the frequency response curve of the MEMS microphone, they considered that the cost of an equalization filter was too high on a low-cost sensor, preferring an *a posteriori* calibration in frequency and level, on the basis of comparison with a reference Class-1 sound level meter. A digital A-weighting filter was however designed. Indoor test measurements were compared with a Class-1 sound level meter showing a very good correlation on the measured sound levels, with a mean difference of 1.6 dB over a 12 mn testing time-period.

In 2018, Peckens et al. [63] have proposed a sensor based on the Teensy 3.6 platform made of a 32-bit ARM Cortex-M4 MCU, on a MK20DX256VLH7 chip, using a XBee-PRO ZigBee Modules (S2B), and an external board for the acoustic measurement (connected to the 16-bit ADC of the MCU, using a sample frequency of 20 kHz). The measurement system was composed of an omnidrectional ECM (PUI POW-1644P-B-R), an amplifier circuit with variable gain and an A-Weighing analog filter circuit. It is interesting to note that the calibration, the A-Weighing filter and the amplifier gain have been implemented using analog circuits, instead of digital processing, in order to reduce the power consumption. Comparisons with sound level measurements using a Class-1 reference device, show a good agreement, with deviations around ±1.5 dB. This system with four nodes and one sink (based on a R-Pi platform), was tested with success in a real environment. The ability of the proposed approach to monitor noise was validated. Some limitations are however highlighted such as insufficient dynamics (50 dB), high residual noise level, as well as power supply issues generating artifacts in the time signal.

In the context of the European Life MONZA project [64], two different low-cost noise sensors have been presented in [65,66], in 2017 and 2019 respectively. The sensors are not clearly described, but it is likely that one is using a digital MEMS (mounted on 1/2 inch support) on a Mini PC platfom, and the other one with an electret microphone (mounted on 1/4 inch support) on a MCU board. Data are transmitted using the GSM 2G/3G protocol and power is supplied by several means: battery, solar panel, electricity network. A digital filter was carried out in order to proceed to the A-weighting filtering, as well as a 1/3 octave band analysis. Ten prototype sensors have been installed in a pilot area in the City of Monza (Italy) since 2017, and seem to provide consistent data in comparison with reference sound level meters, with however a systematic offset around 3 dB. One interesting choice by the authors is the use of digital MEMS, provided an embedded ADC, which limits power consumption and the noise generated by possible interference. However, they observed a reduction of sensitivity of the MEMS during time, requiring to continuously correct the raw data at least until the end of the running-in period of the first few months of use.

In the context of a French national research project (CENSE), Ardouin et al. [67] have presented in 2018 two complementary sensors: firstly, a node using a STMicroelectronics STM32L4 MCU with a ARM Cortex-M4 powered using a solar panel and, secondly, a gateway, based an R-Pi 3 platform, wired powered, acting simultaneously as a sensor and a sink for the nodes. Data transmission between the node and the gateway is performed using the 802.15.4 standard (6LoWPAN MAC layer). The gateways which are located on streetlights are connected to Internet using a wired connection to a GSM router, through a power-line communication (PLC) built using the public lightening system. Acoustic measurements were performed using an omnidirectionnal digital MEMS (Invensense ICS-43432) integrated on a Mini PC, including additional sensors (temperature and humidity). A third octave band analysis and an A-weighting are performed using digital filtering. In addition, the data are encoded to reduce the memory load [26]. The influence of the air temperature have been evaluated in a climatic chamber, showing a very small deviation. However, no details are given concerning the acoustic performances of the sensors.

Recently, an alternative approach was detailed in [68], based on a digital signal processor (DSP) (Texas Instrument TMS320VC5502), to which is connected an acoustic acquisition chain, composed of a Panasonic WM 63-PR electret microphone, an analog conditioning and an ADC (Cirrus Logic CS5344). The processing capabilities of the DSP thus allow the real time calculation of many acoustic indicators as well as a 1/3 or 1/1 octave spectral analysis, simultaneously on two microphone channels. This technical approach opens up many perspectives, due to its high computing capacity in flash memory, which allows the integration of complex additional acoustic processing, with low power consumption. Another special feature of the system is the use of standard interfaces to control the sensor from an external CPU based system, for example a wireless NRG 2 panStamp (with a 868 MHz radio transmission) in the corresponding article. The performance of the sensor, compared to the expected characteristics of the IEC 61672 standard [7], is Class-1 for the DSP part and Class-2 for the whole system, due to the reduced performance of the microphone that was used. Eight sensors were deployed for 3 months in the city of Málaga (Spain), monitored over time and compared to Class-1 sensors. Although the prospects for the used of such a sensor are very interesting, the authors encountered problems of electromagnetic interference, material damage on some sensors, as well as variability due to meteorological phenomena. The authors also recommend the use of MEMS to improve the overall performance of the sensor.

Some other works related to the development of sensor networks for various applications and purposes can also be mentioned, even-though they are not sufficiently detailed to give a complete description. Tan et al. [69] considers a floating-point DSP to perform frequency and time weighting filtering, instead of using the MCU. Botteldooren et al. [21,22] tested noise sensor nodes, based on an ALIX single board computer and an ECM, using the ZigBee radio protocol, to develop sound monitoring networks and algorithms in order to access sound perception. Bell and Galatioto [13] proposed to adapt sensor nodes initially developed for environmental monitoring within the framework of the MESSAGE project [70]. It features 8-bit MCU, a condenser microphone, and an IEEE 802.15.4 data transmission protocol. Fifty nodes were deployed in the cities of Leicester (UK) in order to study road traffic noise. This experiment demonstrated that such technology can be used for urban noise assessment and management. The use of low-cost sensors was also proposed by the City of Barcelona (Spain) as a complementary network of a main Class-1 monitoring network [71].

### 2.3. Synthesis

#### 2.3.1. General Considerations

The study of the literature highlights some key aspects and respective design choices, which are summarized in Table 1 and Table 2. As indicated in the Introduction, our review focuses only on the first element of the chain, namely the sensor, and not on the infrastructure to implement the entire network.

A full comparison of the proposed solutions is unfortunately not feasible for all points of interest. In particular, there is a consistent lack of description of the acoustic performance of the sensors, in terms of residual noise, sound level dynamic or frequency range. Most of the time, the information mentioned generally corresponds to the expected characteristics extracted from the data sheets of the sensor components, which can be assumed to be a higher bound in terms of performance. Indeed, assembled within the sensor with a wide variety of other hardware components and subjected to environmental stress, those performances are likely to be degraded.

Also, different levels of maturity of the proposed low-cost sensors are found. Some developments were limited to the design of the sensors alone (without prototypes), others proposed proof-of-concept (POC) (prototypes and tests), and some others have proposed to deploy several sensors in real urban areas in a quasi-operational framework.

Cost estimates of the sensors given by the authors show that the objective of obtaining a low-cost acoustic sensor (less than 150 EUR) is clearly achieved, with relatively high signal processing capabilities when considering most recent studies [35,66].

The question of the autonomy of wireless noise sensors, while being a fundamental aspect of the design, is in the end not extensively studied in the literature. This is linked in particular to the operating mode of the sensor, such as activity time/sleep time, duration of measurements, number of calculated indicators, and so forth. All those aspects are generally not detailed. However, it should be noted that most of the latest achievements [35,66,67] mention noise sensors directly powered by an electrical network, which seems to illustrate the difficulty of developing wireless sensors with acoustic performances that are relevant to the task at hand.

#### 2.3.2. Sensor Platform

The choice of the sensor platform determines the main functionalities and characteristics of the sensors. Three main families can be distinguished: (1) MCU based existing platforms; (2) specifically developed electronic boards; (3) Mini PC.

The use of existing platforms (1) simplifies the sensor development by using components that have already been optimized in terms of energy consumption (TelosB, CiNet, Teensy USB, Tmote) [33,42,54,56,59,63]. Those platforms generally include all the components needed to develop an environmental sensor (radio communication module, ADC, storage memory, connectors for other sensors, etc.). Such platforms have played a key role in demonstrating the feasibility of developing and deploying noise sensors. In addition, programming the MCU is facilitated by the use of dedicated libraries, but requires a physical connection to a computer.

However, the lack of autonomy and the reduced computing capabilities for calculating relevant acoustic indicators have motivated researchers to develop their own electronic boards [37,38,67]. A specific design makes it possible to select the processor and components best suited to the expected requirements in terms of autonomy and computing power. However, the measured autonomy of the sensors remains to be few days in the best cases, and is very dependent on the periodicity and the duration of the measurements as well as on the data processing that is performed.

The main interest of using a Mini PC, lies mainly in increased computing power [35,59,62,66,67], allowing more advanced digital audio processing, easy integration of external components, of remote updating (without physical connection to a computer), but at the expense of higher energy consumption. In most cases, these sensors are directly connected to the power grid as using a battery limits operation to only a few hours.

#### 2.3.3. Data Transmission Protocol

Choosing a relevant communication protocol for a sensor network normally depends on several parameters, such as the distance between nodes and sinks, data rate, quantity and type of measurements, network topology, global architecture, latency and reliability, cost... Although this problem of choosing a communication protocol and optimizing it has been widely discussed in other areas of sensor network applications, very little has been said about noise sensor networks. The experiments detailed in Table 1 show that different protocols have been used, most often imposed by the choice of the sensor development platform. More information about data transmission protocol are given in Section 3.2.6.

#### 2.3.4. Microphones

As mentioned by several authors [68,73], the microphone is a critical element of the noise sensor. A wrong choice of microphone will impact the quality of the acoustic indicators produced which cannot be overcome. The literature review shows that three types of microphones are considered: (1) electret condenser microphones, and the more recent MEMS microphones that can be either (2) analog or (3) digital. The replacement of ECM microphones using MEMS ones was justified by the authors on the basis of their acoustic performances that were *a priori* more interesting for their use in acoustic measurements [56,62]. In addition, MEMS microphones have reduced dimensions, are relatively reliable and durable, and above all are produced at a lower cost. The use of MEMS therefore was considered particularly relevant in the context of the implementation of low-cost sensor networks, specifically for urban noise monitoring. To our knowledge, the first work mentioning the potential interest of MEMS for applications in acoustic measurements in urban environments, can be attributed to the National Physical Laboratory (NPL) in the UK [45] in 2004, which will later lead to the DREAMSys sensor prototype. While the overall results are rather positive, some authors have highlighted some limitations and problems that are mainly related to the technology and layout of MEMS used in sensors. In order to fully understand these limitations, the following paragraph provides a brief description of MEMS microphone technology.

A MEMS microphone is composed of a sensor (MEMS sensor) and integrated circuits. The MEMS sensor is a silicon capacitor made of two electrically isolated surfaces. One surface, called the backplate is fixed, and is covered by an electrode and made of many holes, that is, acoustic holes. The other is movable and is called the membrane or the diaphragm. Sound wave passing through the acoustic holes of the backplate will set the diaphragm in motion, creating a change of the capacitance between the two corresponding surfaces, which is converted in an electrical signal by the application specific integrated circuit (ASIC). The ASIC delivers an analog output or a digital output, depending of the microphone type (analog or digital). The MEMS microphone and the ASIC are packaged together, surrounded by a substrate and a lid, forming a cavity. A sound inlet (acoustic port) is present either in the substrate (bottom port configuration) or in the lid (top port configuration), and, most of time, positioned directly in the MEMS cavity. For analog MEMS microphones, the electrical output signal from the ASIC is sent to an external pre-amplifier, also in charge of converting the output to a signal that can be used as input of an acoustic chain. For digital MEMS microphones, the ASIC output is sent to an internal analog-to-digital converter (ADC) to provide a digital signal, either as a pulse density modulated PDM format (1-bit high sample rate data stream) or I2S format (same as PDM microphone but including a decimation filter and a serial port in order to produce a standard audio sample rate).

Acoustic performance of MEMS depends on mainly technical aspects, such as the location of the sound inlet [74]. Considering the bottom configuration where the sound inlet is above the cavity (front chamber), the interaction between the air in the sound inlet (back chamber) and the air in the MEMS front chamber creates an Helmholtz resonance, whose frequency increases as the air volume decreases, positioning the resonant frequency at the upper part of the frequency band of interest, and thus, leading to a flatter frequency response. Additionally, locating the sound inlet above the MEMS sensor allows easier interaction between the diaphragm and the sound wave, thus increasing the sensitivity and the SNR. Conversely, considering the sound inlet at the top will create a larger air volume in the cavity between the lid and the backplate (front chamber) in comparison with the air volume in the cavity between the diaphragm and the substrate (back chamber); it generates a lower resonant frequency, thus possible resonant frequency in the frequency range of interest. Such unwanted effects were observed, for example, in [45,62]. In addition, due to a smaller volume in the back chamber for the top configuration in comparison with the bottom configuration, it will be more difficult for the diaphragm to move, leading to worst sensitivity and SNR. Thus, it appears that the bottom configuration provides better acoustic performance.

The thickness of the PCB on which the microphone is soldered can also modify the volume of the front chamber and the inlet length altering the upper frequency response of the microphone [38,45]. This effect can also be more pronounced if a cover is added on the device, as a protection [75]. A possible solution is to mount the MEMS and the ASIC in the inner side of the lid, for the top configuration, leading to the same expected performance than for the bottom configuration. Barham and Goldsmith [45] also mentioned that the background noise of MEMS microphones was significantly higher than classical microphones. In practice, experiments described in the literature mention noise levels between 20 and 30 dB, depending on the frequency [76]. However, recent works on the development of new generation MEMS microphones suggest the possibility of developing noise sensors with high acoustic performance [77], which makes it the ideal component in the future for the development of noise high-performance sensors.

#### 2.3.5. Frequency Weighting

Most acoustic indicators for the assessment of environmental noise require frequency weighting (generally A-weighting) to take into account the sensitivity of the human ear to certain frequencies. Since the calculation of acoustic indicators, such as equivalent sound levels, is integrated within the sensors, this weighting should be done as a pre-processing, using analog filtering [45,49,63] or digital filtering [38,61,62,66,67,69]. Analog filtering makes it possible to overcome the computing limitations of the microprocessor used, reduces energy consumption, can be processed in real time and avoids any bias, such as rounding errors during digital computation [78]. However, analog filtering of the audio signal constrains the nature of the acoustic indicators at the output of the sensors, unless the filter is replaced. Conversely, digital filtering offers more flexibility, but at the cost of reducing the possibility of real time computation if the microprocessor is not powerful enough, and potentially reducing autonomy [63]. Risojevic et al. [38] recently compared several techniques for digital A-weighting filtering and showed that using a matched z-transformation filter with a cascade form implementation was relevant for a small processor core, and slightly better than a bilinear transformation [67], when comparing with an analog filter.

#### 2.3.6. Frequency Equalization

The acoustic acquisition chain generally has a frequency response curve that is not as “flat” as expected, often due to the frequency characteristics of the microphone. This creates a bias on the measured audio signal, and therefore on the calculated acoustic indicators. Some authors therefore propose to compensate the frequency response of the acquisition chain by implementing an equalization filter [61,62]. However, this filtering requires a significant workload on the microprocessor, with an impact on the sensor autonomy. Another solution proposed by [48], implemented in their own MEMS design in stainless steel tube, was to use a patented acoustic filter made of a resistive element and a closed volume, the whole system acting as a low-pass filter.

Low-cost microphones can exhibit significant variability in frequency response. An equalization filter should therefore be determined for each microphone, which would be ideal, but is not possible when deploying a large number of sensors. Mydlarz et al. [62] thus propose to generate the equalization filter on the basis of an average of a limited number of microphone impulse responses. The authors observe a good behaviour of the implemented real-time time-domain filter (based on an inverse linear-phase finite impulse response filter), however with an increasing of residual noise between 20 Hz and 400 Hz.

#### 2.3.7. Calibration

At the output of sensors, sound level indicators must be adjusted to give measures that are as close as possible to sound levels measured by a reference device, such as an acoustic calibrator. In the best case, this adjustment takes into account the variability of the microphone sensitivity, but this adjustment can also correct linearity defects in the acquisition system or in the digital processing chain. Two calibration methods are considered in the literature, either by using an acoustic calibrator (mainly 94 dB at 1000 Hz) [59,66], when the microphone mounting device allows it (matching the diameter of the microphone mount with the acoustic calibrator), or by comparing with a reference sound level meter under the same measuring conditions [49,57,62,63]. In the simplest case, the same correction is applied to the entire temporal signal, without distinction of frequency or amplitude. In a more advanced way, this correction can also correct linearity defects in level and frequency [38] and temperature [73]. The correction can be taken into account either within the sensor or in post-processing once the data has been collected on a server. In most cases, this correction is applied to the digital signal, but it can also be integrated into an analog circuit [63].

#### 2.3.8. Noise Indicators

The choice of output acoustic indicators is very important for designing the sensors in terms of expected computational and power resources.

A temporal integration in order to obtain an equivalent sound level over a given integration time (1 s for example), will require far fewer resources (energy and calculation) than the calculation of frequency band spectra. Such time integration can easily be processed by a system based on an MCU with battery, while a frequency analysis will require more resources, as proposed today by a mini PC, with a wired power supply [35,62]. Most authors have therefore limited themselves to produce equivalent sound levels, generally with an integration time of 125 ms, that is, fast time weighting, and 1 s, that is, slow time weighting. However, with the help of technological developments, it can be seen that some authors put forward solutions that *a priori* make it possible to carry out frequency band analysis with an MCU [66,67].

While the measurement duration is often indicated in the literature, the temporal periodicity is rarely specified. It is also quite difficult to determine whether the calculations are carried out continuously and in real time, or over periods interrupted to calculate noise indicators, to transmit data and to save power resources during sleep periods.

The transmission of audio signals is clearly not a priority, for obvious technical reasons: network bandwidth, storage on sensors and servers, and energy consumption but more importantly for privacy concerns. The calculation of acoustic indicators directly within the sensors, and then their transmission, is the only relevant solution. However, one can cite a few exceptions—Mydlarz et al. [35] propose to store and transmit audio samples, but only for testing purposes to validate machine learning algorithms for identifying sound sources. Similarly, Sevillano et al. [72] recorded audio data by connecting the sensor to a digital recorder, in order to test an acoustic event detection algorithm. In both cases, the final objective is to integrate, after optimization, these algorithms directly into the sensors. On the other hand, biodiversity monitoring rather emphasizes the need to preserve raw audio signals, which creates constraints for continuously capturing and transmitting data in a fully autonomous wireless SN [31,33,79].

#### 2.3.9. Meteorological and Outdoor Conditions Effects

Renterghem et al. [73] have studied the effect of temperature, humidity and wind, on the sound levels measured by electret and MEMS microphones. It was shown that applying an air temperature correction may have a positive effect on the long-term measurements. For example, by comparison with a reference microphone, such temperature correction (acting as a gain on the signal) applied on an ECM was able to reduce the deviation of the global error from 1.6 dBA to 0.8 dBA, over the full period of observation (several months). In order to proceed to such correction, the authors proposed to consider an additional sensor for measuring the temperature. Because relative humidity and temperature are highly correlated, the temperature correction may also include the humidity correction. The authors have also noted inconsistent behaviour of the MEMS microphone at temperatures below 20 °C, high relative humidity and high wind speed, but no explanation has been given.

The effect of ambient temperature on the sensitivity of a MEMS sensor was also investigated by Barham and Goldsmith [45]. From their results, it seems that the variation in sensitivity would increase with frequency and temperature (tested between −5 °C–40 °C), in the order of ±1 dB over the frequency range between 100 Hz and 8 kHz. Conversely, [67] have not seen significant variation of MEMS microphone sensibility with temperature.

In another study, Bartalucci et al. [80] report that they observed a reduction of acoustic sensitivity of MEMS microphones during the initial running-in phase, on the order of 1–2 dB on a time period of 4 months.

A procedure was proposed in [76] in order to evaluate the evolution of acoustic performances of MEMS and electret microphones, when exposing to stressing conditions. Microphones were installed inside a salt spray chamber in order to exaggerate the possible damage in outdoor environments (such extreme conditions can however not be reproduced in real environments). Authors observe a slightly better stability for MEMS microphone in term of frequency responses and noise floor. However, anomalies like spike occurrences have been observed in the audio signal measured by some MEMS microphones, whose impact is low in terms of equivalent sound level, but which was remained totally unexplained.

Li et al. [81] have studied the reliability of MEMS microphones, through accelerated life tests in a corrosion test chamber and in a pneumatic shock testing. The second testing is more related to the use of a MEMS microphone in smartphones, and has showed that failures of the diaphragm and backplate of the MEMS can be observed, but only when an impact is generated in the direction normal to the diaphragm and for very high acceleration level (greater than those observed in real life). Considering the corrosion test, wire bond corrosion and membrane embrittlement were observed after 90 days in the test chamber, but with a very slight impact on the frequency response of the microphone.

All the results mentioned above should be considered as points of attention, and not as “absolute” results, since the observed effects may vary due to many design factors. The generation, the type of microphone, but also the presence or not of wind and rain protection devices, all will have an impact on the experimental results. In our opinion, the conclusion to be drawn is that it seems mandatory to carry out specific tests, before the sensors deployment, in a climatic chamber (as also suggested in [62]) or a corrosion chamber for example, in order to evaluate the temperature effects on the sensor, to assess whether a correction should be applied. In addition, protection of the overall sensor (not only the microphone) against wind, chemical agents, dusts, pollutants, shock... seems essential, and their impact must also be evaluated [76,81].

## 3. Noise Sensor Design for Low-Cost Networks

The literature review highlights the rapid evolution of low-cost sensors, mainly driven by the more global need for electronic and computer systems dedicated to the development of smart sensors. The technical solutions available today seem both sufficiently stabilized and adapted to offer low-cost acoustic sensors that can meet current and future applications for the assessment and management of environmental noise.

In this section, on the basis of the literature review detailed previously, we propose to highlight the technical characteristics that we believe are essential for the correct design of a low-cost acoustic sensors, offering a level of performance in line with the needs in noise monitoring. Depending on the objectives and applications considered, two levels of expected sensors characteristics can be distinguished, ranging from minimal to optimal (Table 3). It is these characteristics that condition the technical elements to be considered for the realisation of the noise sensors.

### 3.1. Expected Characteristics of Noise Sensors

#### 3.1.1. Acoustic Measurement Accuracy

Referring to conventional measuring systems and historical practices in environmental acoustics, one could try to compare the acoustic performance of low-cost sensors with Class-1 or Class-2 devices for ’expertise’ or ’control’ environmental noise measurements [7]. However, as pointed by [46], there are situations where Class-1 or even Class-2 systems give measurement results whose very high accuracy is not necessarily in line with the practical use made of these data. This is particularly the case in the strictest application of the European Directive 2002/49/EC on the assessment of environmental noise. Informing the public through "coarse" noise maps, such as the establishment of action plans to reduce noise pollution, does not require a very high precision in the performance of acoustic measurements.

This is also highlighted in [22], mentioning that low-cost acoustic sensor characteristics are already quite consistent, in term of metrological capabilities, with what is expected from a strategic noise map: it is not necessary to seek a measurement accuracy of less than 1 dBA; the sound spectrum is rather centred around 1000 Hz, that is, over a frequency range for which a low-cost sensor is not lacking; a rather high minimum noise level, therefore beyond the residual noise of the sensor (except, maybe, for quiet areas).

For more specific needs, the most important guideline is to be aware with the technical limitations of the systems developed, and to ensure that the exploitation of the collected data is consistent with these limitations.

#### 3.1.2. Acoustic Indicators

Indicators of interest to describe urban noise environments are calculated on the basis of 1 s or 125 ms data. They cover at least equivalent sound levels as well as statistical indicators, and, sometimes, emergence indicators such as the number of exceedances at given thresholds. Finally, some authors have introduced more demanding indicators for specific uses, such as the time and frequency second derivative (TFSD) [82] to describe voice and bird sounds, which requires a good temporal and spectral response of the sensors. Consequently, the expected characteristics of noise sensors are guided by the amplitude of the sound levels encountered and the spectral information of the sources to be characterized. The characterization of urban noise environments, from quiet areas to the noisiest events, assumes a linear sensor response in a range from 30 to 105 dB(A). The interest in bio-phonic sources, especially bird sounds, suggests that we should also be able to accurately measure high frequencies, up to 16 kHz.

### 3.2. Sensor Platform and Components

The choice of the platforms and components for the sensor development is mainly determined by the questions of how the sensor is connected to the network, how the sensor is powered, and what are the expected sensor output indicators. This last question also conditions the first two questions.

A sensor that transmits data by radio will be limited by the maximum data rate of the transmission protocol, as well as the distance and visibility to the nearest gateways or relays. The power supply mode will determine the computational and storage features of the sensor, as well as the operating conditions depending on the energy recovery mode (battery change or power supply using renewable energy).

Several technical solutions can be considered, each with different components/functionalities—a sensor connected with a wired connection to the electrical network and to the data network; a sensor powered to the electrical network through a wired connection, but transmitting data by radio wave; an autonomous energy sensor (possibly also acting as a relay) transmitting data by radio wave.

#### 3.2.1. Wired Sensor Platform

With regard to the experiments presented in the previous literature review, the choice of a Mini PC constitutes an optimal choice with regard to the low-cost, the computing and storage capacities, the connectivity with other modules (radio, other sensors...), the remote maintenance and update of the system, the change of some modules (since not all modules are integrated into the motherboard of the Mini PC but just connected). There are several Mini PC solutions available at a very affordable price, with fairly similar features, and with different operating systems. Among these, it is clear that the R-Pi family seems an excellent choice given the many accessories and modules available, but also given the presence of a very active community. The latest R-Pi models use very powerful 64-bit microprocessors, but at the expense of higher power consumption. The choice can be made for an older model (model A+ or Zero), very cheaper, but with a better power/energy consumption ratio if it were to be run on battery power [83].

#### 3.2.2. Wireless Sensor Platform

The development of a stand-alone sensor is more complex as it must meet many requirements. The nature of the acoustic indicators to be produced (continuous sound levels and spectra) requires a powerful microprocessor; the dynamics of the sound levels to be measured requires quantification by the ADC of at least 16 bits (96 dB of dynamics), ideally 24 bits to take advantage of a wider dynamic range (144 dB), which needs the use of a 24 or 32 bits MCU, as in the STM32 series already used by several authors [38,67].

#### 3.2.3. Microphone and ADC

The acquisition chain (microphone, gain amplifier, ADC) is the other essential element to consider. Most of the achievements have focused either on ECMs or on MEMS, combined with an external ADC. Feedback from the literature review has shown the sensitivity of the acoustic signal to electrical and radio frequency interference, causing an increase in the residual noise. This is why some authors have recently turned to digital MEMS (the analog-to-digital conversion is performed inside the microphone) [35,66,67], which seems today an optimal choice. In addition, using a digital MEMS microphone with an I2S interface, it is unnecessary to use an external codec [84].

The choice of sampling frequency depends mainly on the spectral band of analysis, which depends on the expected sensor application. The optimum corresponds to the audible frequency band 20–20k Hz, covered by most MEMS microphones, which implies a standard sampling frequency of 44.1 or 48 kHz. While such a sampling frequency is not a problem for a wired sensor (such as a R-Pi), it is more problematic with MCU, and even more if a real-time processing is required. Currently, the only reference for a stand-alone node with a digital MEMS node mentions a sampling frequency of 32 kHz [67], but the paper does not described the final sensor performances.

#### 3.2.4. Noise Floor Enhancement

As mentioned by several authors, residual noise is one of the elements of the measurement chain that can limit a sensor ability to perform measurements at low levels, as in quiet spaces. The observed residual background noise levels are generally much higher than the value indicated by the manufacturer for the microphone and are caused, for example, by interference on analog electronic circuits or by the limited performance of some ADCs. The solution to reduce the residual noise is to optimize the electronic components of the sensors. Another way, proposed for example in [73], is to combine several microphones on a same sensor and to reduce the residual noise by applying a noise reduction method based on cross-correlation techniques. Such procedures may be however power and computational consuming. It must be noted that this use of several microphones simultaneously to reduce background noise would also make it possible to envisage other applications of this type of acoustic sensor, such as locating sound sources.

#### 3.2.5. Mass Storage

Sensors can also have mass storage capacities to store various information, for sensor maintenance, but also to temporarily store the collected data when the connection to the gateway or data server is interrupted. The sizing of this memory must take into account the duration of temporary data backup, and potentially the ability of the sensor to transmit a large amount of data once the connection is established, while simultaneously collecting and processing new data.

The type of storage device is the second element to be considered. The more relevant choice is to use a flash memory (memory card, USB flash device, SSD), offering lower power consumption, easier maintenance, higher transfer speed, no operating noise, but at the expense of less mass storage than a traditional hard disk (HD), but also shorter lifetime due to a limited number of write cycles and higher cost.

#### 3.2.6. Data Transmission Protocol

Depending on the type of output noise indicators (such as 1/3 octave band analysis [35,66,67] or audio capture [35]), measurements may require a very high frequency, inducing a huge quantity of data to be transmitted. Conversely, a temporal integration of the audio signal does not generate an enormous amount of data to be transmitted.

With regard to the applications that are currently envisaged for these noise sensors (see Introduction) and the expected output indicators (Section 3.1.2), the needs in terms of data rate are rather increasing. The targeted useful data rate should be up to 10 kb/s with a maximum range around 100 m to be able to transmit compressed data from one node to a sink, in a urban configuration, taking into account trees, buildings, cars and trucks impacting the radio signal propagation.

The use of a wired network is obviously the simplest and most effective solution to ensure data transfer under ideal conditions. This solution will be probably feasible in the future, given the growing number of cities developing smart and connected systems. Nevertheless, the expected spatial density of noise sensors requires dating the use of radio transmission, as most past experiments have envisaged.

Considering the amount of data to be transmitted from a noise sensor, the implementation of a very simple topology, that is, a direct transmission from a node to a sink, seems the most obvious solution. Topologies involving one or more relays do not seem to be possible today, given the technical solutions that are available. In the literature, one recent reference [67] mentions the use of sensors that also act as relays for other sensors, but this has not been developed further.

As a first solution, the 3G/4G and other GSM protocols could be considered. However, such transmission protocols are not cost effective for very dense sensor networks because with one subscriber identity module (SIM) card is required per sensor (or two SIM cards in order to ensure the relay of data transmission in the event of a GSM network failure). Regarding other data transmission protocols without subscription, several solutions are possible, depending of the range/data rate compromise that is expected for the sensors networks (Table 4).

Looking at the available data rates, the LPWAN technologies like LoRaWAN and Sigfox would not allow data transmissions with a sufficient efficiency. Even for the LoRaWAN, the maximum data rate is higher than the useful data rate due to the overhead of the protocol. In terms of data rate, the Wi-Fi and the Bluetooth protocols would be more efficient, but the battery life will be too limited for an application without constant energy and for an efficient coverage of an urban area. Zigbee and 6LoWPAN, based on 802.15.4 specification, present both maximum range and data rates that are compatible with the noise SN.

#### 3.2.7. Additional Sensors

As mentioned in Section 2.3.9, the knowledge of the air temperature can be sometimes useful to calibrate noise sensors. More generally, knowledge of atmospheric and meteorological conditions can be interesting for a better use of data. For example, the presence of rain or a strong wind can cause disturbances to the measured acoustic signal, which, if not identified, can lead to misinterpretation of the collected data. The measurement of these atmospheric conditions (temperature, humidity, wind speed and direction) at the same time as the acoustic signal seems relevant, particularly because of the low cost of the components and of the limited additional data it can generate. More generally, the possibility of connecting other types of sensors (traffic, ambient light, air pollution, video, accelerometer, etc.) [38,49,60] would make it possible to develop a global and multi-disciplinary environmental approach [9,70] by pooling technical resources. The integration of multiple sensors adds additional constraints in terms of maintenance, data storage and transmission, as well as energy consumption, which must be anticipated.

### 3.3. Sensor Life

One important issue is to determine the expected lifetime of a low-cost sensor. Knowing that the lifetime of a Class-1 sound level meter can extend to more than 10 years under normal conditions of use [90], a lifetime of a few years (typically 5 years) already seems an ambitious goal considering the overall cost of a low-cost noise sensor and the quality of its internal components.

There are several components that can affect the lifetime of a sensor, mainly the measuring microphone, the data storage elements, and, if applicable, the battery. All other electronic components embedded in a sensors are designed most of time to operate for more than 10 years without problems in outdoor conditions, except when experiencing unplanned event, such as mechanical damage, high level of humidity, or really extreme temperature conditions out of the expected −20/+55 °C range.

As detailed above, recent microphones and especially MEMS have a fairly good resistance to atmospheric conditions and have a limited drift over time. Newer mass storage devices also have a longer life expectancy given today’s permitted read/write cycles. Finally, as far as autonomous sensors are concerned, the most sensitive component is undoubtedly the battery. Either the battery is removable, in which case an on-site intervention is required, or the battery is rechargeable and in this case the life cycle is defined by its ability to recharge, generally by using solar panels, while maintaining optimal properties.

Most solar panel lifespan is around 20 years, with a power output decrease of less than 1% per year [91]. The sensor will then be given around 80% of the initial energy after this time. Environmental conditions, will have also an impact on the longevity of lithium batteries (i.e., the type of battery most commonly used in electronic devices): the worst case is for high temperature (above 40 °C). Most of time, battery packs do not die suddenly, but the runtime gradually shortens as the capacity fades. The capacity of the battery will also decrease during its life, starting from 95% of its nominal capacity it quickly decreases to around 80% in less than a year (with 250 charge/discharge cycles). In addition, the depth of discharge will have also an impact of the battery durability: considering smaller discharges will prolongs the battery life. The lifetime of such batteries can typically range from a few years (typically 5 years) for consumer products to more than 10 years for industrial products.

### 3.4. Power Resources

As pointed out in [54,56], autonomous nodes, made with a battery and a solar panel, should present enough storage capacity to store the energy that is required over the duration of the measurement. For low cost sensors, the dimensions of the solar panel as well as the dimensions of the battery are a tradeoff between the energy consumption of the sensor and the overall price of the sensor.

Considering an average power consumption of the sensor around 75 mWh, which seems sufficient for a MCU based noise sensor already offering significant computing power, the solar panel should be able to provide enough energy to power 24 hours of energy request, even during the worst month of the year (not in extreme conditions). If during this period, only 3 hours of sun are available, a solar panel should provide 24×0.075=1.8 W, which seems very reasonable in terms of cost and space requirements. To be able to power the sensor during a few days without sun, the battery should have the biggest possible capacity. A 2600 mAh battery will be able to provide enough energy to a sensor during a few days, even if there is no sun available to recharge the battery.

It is also essential to consider the progressive degradation of the properties of the solar panels and batteries in order to ensure the correct operation of the sensor over the envisaged lifetime. From the initial design stage, this means overestimating the capacities of the panels and batteries to ensure trouble-free operation of the sensors over the expected service life. Lastly, one can also mention that the technical improvement of the energy system must also be accompanied by the development of algorithms and procedures to optimize or reduce energy consumption [33].

### 3.5. Acoustic Calibration

Regardless of the intrinsic performance of the sensors, calibration is an essential operation for any “controlled” acoustic measurement. At a minimum, the sensor calibration should be performed, using an acoustic calibrator, for example, 94 dB at 1000 Hz, to determine the sensitivity correction at the selected frequency, under controlled conditions. The use of a multi-frequency calibrator can be useful in determining a frequency correction, unless the frequency response has been corrected using an equalization filter within the sensor. Considering the expected accuracy of a low-cost sensor, the use of a Class-2 calibrator seems sufficient. The use of a sound calibrator requires that the microphone be mounted on a cylindrical support with a compatible diameter.

As pointed out in [66], it is important to regularly check this sensitivity correction throughout the period of use in order, if necessary, to take into account the variations in sensitivity of the measuring microphone. Taking into account the sensitivity correction directly within the sensor, in pre-processing, (rather than retrospectively on the post-processing data server) seems relevant to ensure that the acoustic indicators at the sensor output are consistent with reality. However, even in this case, the verification procedure with a sound calibrator is required. Such correction was for example proposed in [63], using an electronic circuit, but the it can also be carried out by applying a correction on the raw data.

Because of the variability between low-cost microphones, particularly in terms of frequency response, it may be tempting to determine, in laboratory, a sensitivity correction for each sensor. However, this can quickly become tedious to do for a large number of sensors. It seems more appropriate to estimate a correction value on a sample of sensors, then calculate an average correction that will be applied to all sensors, as proposed in [62].

Moreover, the question of the calibration of a very large number of sensors, particularly in situ, is a very hot subject of research [92] that could be considered to noise sensor networks, for example by developing automatic calibration of sensors without human intervention.

### 3.6. Additional Challenges

The development of a low-cost acoustic sensor for long-term acoustic measurement is only the first step in a comprehensive approach to the development of a sensor network for noise monitoring. Many other aspects, such as the development of an optimal technical and IT infrastructure for network and data management, anomaly detection, optimization of sensor positions, spatial and temporal data sampling, and the management of hybrid networks, are important challenges, which are still relatively open in the field of noise monitoring as of today. The complete review of all those issues that surround the sole design of the sensor is left for future work. However, for the reader to grasp some aspects of those challenges, we present in the following some related advances and discussions.

#### 3.6.1. Detecting Network Defaults

It cannot be expected from a low-cost sensor the same performance as a professional sensor in terms of reliability and durability. Therefore, and as has been mentioned in several studies [50,73,93], the probability of malfunctioning of a low-cost sensor must be carefully considered. If extreme cases (for example, due to hardware malfunctions, a sensor no longer returns data) can be immediately detected, others irregularities (such as abnormal acoustic data behaviour due to certain weather conditions [73] or sensor power loss problem [63] for example) are more difficult to identify. The implementation of advanced algorithms for dysfunction detection is therefore essential [35,73]. The subject of automatic fault and anomalies detection is particularly developed in the literature about wireless SN, but only little for noise monitoring. The description of these methods is outside the scope of this study; the reader may refer to recent references about this subject [94,95,96,97].

#### 3.6.2. Temporal Sparse Sampling Strategies

The question of the duration and frequency of acoustic measurement is crucial since it concerns different aspects of the definition of sensor characteristics, such as memory space for data storage, computing capacities for real-time processing an data transmission rate. Reducing the measurement time therefore makes it possible to be less demanding on the characteristics of the sensor, and thus to reduce its cost and to increase its lifetime. Thus, it may be particularly interesting to study the temporal structures of noise levels in the environment as well as their spatial dependencies, in order to potentially reduce the sampling duration.

As an example, Can et al. [98] investigated the variations of hourly noise levels at the week scale, showing that a matrix of relationships between hourly noise levels can be defined for each measurement point, from which noise levels at any period can be deduced from measurements at other periods. The temporal trends are however different depending on the site. A method for stratifying urban space has also been proposed in [99]. Four categories were considered were arterial roads outside the central zone, arterial roads in the central zone, two-way roads connecting different zones, and one-way roads. The interest of such stratification lies in the fact that temporal variations in noise levels are correlated from one point to another within the same category.

However, these studies are still insufficient to optimize the sensors in terms of their temporal measurement dynamics and should be continued.

#### 3.6.3. Optimizing Sensor Locations and Network Deployment

##### (1) Spatial Representativeness and Interpolation

Even if the very low intrinsic cost of individual sensors may encourage to not limit the number of measurement points to be integrated into the network, in practice, it must be limited to reduce the overall cost of the networks, mainly in terms of maintenance. Thus, dealing with a limited number of sensors, the choice of the ’best’ location of each sensor may be of major importance, since many locations are possible in a large urban areas. In addition, authors have also questioned the spatial representativeness of noise, in order to limit the number of sensors, and then to consider ways to interpolate noise indicators between the measurement locations.

Can et al. [100] showed that interpolation methods were defective when the spacing between sensors was too large (about one measurement point every 250 m in the study). The explanation given is that they do not offer a sufficient covering of the network, and assume spatial variations that are not coherent with traffic dynamics or street configurations. Indeed, in urban areas, a distance of 250 m can see a succession of very varied environments.

The study of the spatial characteristics of sound environment variations help defining interpolation functions. Gozalo et al. [101] showed that a stratification of roads based on their functionality was helpful before interpolating sound levels showed similarly, based on a measurement campaign in the city of Plasencia (Spain), that the characteristics of sound level variations follow the categories formed with road functionalities.

Liu et al. [102] analyzed the sound environments of the city of Rostock, Germany, and observed that spatial variation of urban soundscape patterns was explained by underlying landscape characteristics, while temporal variation was mainly driven by urban activities.

Zuo et al. [103], based on measurements in the city of Toronto (Canada), observed that noise variability was predominantly spatial in nature, rather than temporal: spatial variability accounted for 60% of the total observed variations in traffic noise.

Two examples of spatial interpolation of noise levels based on a dense sensor network can be found in the literature. In [60], a fix grid of 78 sensors was deployed in the city of Algemes (Spain). The network covered 1.8 km${}^{2}$, which is about a square grid of 50 m on each side. For the purposes of the study, 10 sensors were removed in which levels were estimated at five 3-hours periods of the day by an interpolation method, namely an Ordinary Kriging in which noise levels are described by a logarithmic function. The study shows that under this sensors density the kriging method seems an efficient method to interpolate noise levels, within a RMSE of 3.5 dB(A). In addition, the residuals are spatially correlated except for the [19 h–22 h] period, probably because it entails specific noise behaviors (leisure noise activities, etc.).

In [5], the impact of the density of observation points and the performance of four spatial interpolation methods were presented. Mobile measurements have been performed while walking multiple times in every street of the XIIIrd district of Paris (France), to construct a reference map, which is estimated by adaptively constructing a noise map based on these measurements. The four interpolation methods were constructed by combining two algorithms: (i) the Kriging method, either Ordinary Kriging or Universal Kriging (which consists in adding a linear trend, defined from the distance between each location and its closest road in each amongst four categories) and (ii) the definition of the distance between locations, either Euclidian or computed from the road network.

##### (2) Best Sensor Location

Huang et al. [104] proposed a hybrid model, based on a K-means clustering algorithm and an immune technology particle swarm optimization algorithm, to define the best locations for measurements stations. The methodology was applied to a real urban areas, showing that 28 optimization measurement points could replace the original 100 grid noise survey points.

In the more general context of sensor networks for environmental monitoring, Reis et al. [105] suggest that models can also be used to optimize the deployment of sensors by identifying areas of interest according to specific metrics. Applied to the noise monitoring field, noise modelling could be used, for example, to detect whether locations in a set of potential measurement points are highly correlated, which in this case would reduce the number of measurement points.

Beyond the metric itself associated with the measurement (i.e., the acoustic measurement in the present case), other elements can also be considered in the deployment and of the optimization of the whole network, such as the connectivity between node/relay/sink, the spatial domain coverage or the network life and energy efficiency [106,107].

#### 3.6.4. Considering Hybrid Networks

Accessing to complementary noise data, that is, data produced by other sensors, can be useful to increase the relevancy of end-user applications.

A low-cost SN can used for example to complement an existing professional quality network (i.e., using Class-1 measurement systems). Data can be shared within the same database and used simultaneously to further evaluate sound environments. This is, for example, the choice of the city of Barcelona in Spain [71]. This makes it possible to extend the spatial coverage of observation at a lower cost. Provided that the data from two networks are in a compatible format, the processing and analysis of the data from the low-cost sensor network can thus take advantage of the existing tools for the professional network, which again limits the cost of the investment.

Merging noise data produced by smartphones in the framework of a crowd-sourcing noise approach [108], with data obtained using a low-cost SN, can increase the quality of the soundscape evaluation. In opposition to ’static’ sensors, smartphones can be seen as ’mobile’ sensors. In a complementary way, noise sensors located on cars, bus or bicycles [109] are also considered as mobile sensors. Such sensors could be used to increase the relevance of the sound environment database and to reduce the cost of the network. However, considering both mobile and static sensors in a noise network can introduce significant challenges in terms of mobile sensors detection/localization and data transfer to a static gateway [110].

This hybrid network approach can also be extended to networks dealing with multiple pollution. Noise sensors, at the lower cost of environmental pollutant sensors, can then be used as a proxy to estimate airborne pollutant concentrations or number of fine or ultra-fine particles with a limited number of sensors [111,112]. One of the ambitions of these multidimensional treatments is to establish possible confounders in the characterization of the health impacts of noise and air pollutants [113], or noise and fine and ultra-fine particles [114], as expected by epidemiologists. The use of acoustic indicators as proxies for estimating air pollution quantities, motivated by the lower cost of acoustic sensors, faces however many obstacles [115], including a different dispersion behavior that leads to very variable correlation coefficients.

## 4. Conclusions

Given the major problem of evaluating and controlling sound environments, the development of low-cost sensor networks is today an interesting alternative solution, complementary, to more traditional solutions such as modelling and "professional" observation networks. Numerous researchers have thus focused on the development of low-cost sensors over the last fifteen years, ranging from proof-of-concept to the deployment of operational networks [29].

From a technical point of view, Table 1 illustrates fairly well the evolution of low-cost sensors, from the adaptation of existing sensors (but with limited resources) to the use of mini-PCs and MCUs (with more extensive computing and measurement capabilities). If the sensors can be directly powered by an existing electrical network, mini-PCs are the most relevant solution up-today, especially in view of the modularity and real-time processing capabilities they offer. For stand-alone sensors, most recent MCUs offer interesting performances, but their overall capacities remain very dependent on their power supply and recharging mode. From an acoustic measurement point of view, the use of a digital MEMS with a sampling frequency of 44.1 kHz now seems to be a technically affordable solution, not very sensitive to electrical and electromagnetic interference, that meets the challenges of noise monitoring. Among the possible technical evolution, the development of sensors composed of several microphones would offer new perspectives for the localization and the tracking of sound sources, as well as for measuring 3D audio [30,68,116]. Concerning radio data transmission, Zigbee and 6LoWPAN protocols, based on 802.15.4 specification, present both maximum range and data rates that are compatible with noise measurements. Setting aside the problems of sealing against weather and pollutants, as well as the mechanical protection of the sensor, which can be solved by integrating the electronic components in a specially designed container, the service life of the electronic components, including memory and battery, is now potentially fully compatible with long-term acoustic measurement.

The individual cost of a sensor must be put in relation to the overall cost of an infrastructure consisting of a very large number of sensors [93], potentially requiring a high level of maintenance. The question of the best location of sensors is therefore an important issue for the future. In addition, the automatic detection of anomalies in the network, whether to identify a hardware malfunction or an abnormal set of data, are also subjects that will have to be addressed to improve the database quality. The multitude of such data also raises the question of developing appropriate data infrastructures for their representation and processing [35,48].

There are many opportunities that enhance the value of these measurement networks and the collected data. Environmental services of cities can, for example, use data to dynamically adapt their policies, since they are able to measure directly the effects of the policies tested. Another example is that local residents associations, with the help of specialized services, can understand the environmental quality of their neighborhood and use it to alert the authorities or become a source of proposals. To do this, it is also important that current networks be enriched with perceptive data, in order to better describe the impacts of noise on citizens.

## Figures and Tables

**Figure 1 sensors-20-02256-f001:**
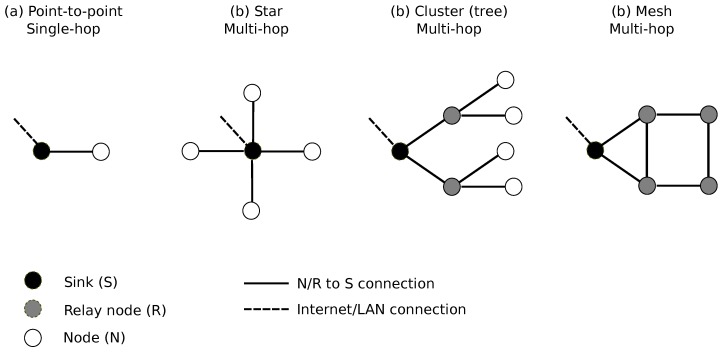
Sensor network definitions and topology examples.

**Table 1 sensors-20-02256-t001:** Main characteristics of low-cost noise sensors detailed in the literature: node based development platform, quantization level of the MCU, ’N-2-S’ node-to-sink data transmission protocol, ’Mic’ microphone type (ECM, analog MEMS (a-M), digital MEMS (d-M)), ADC quantization and sample frequency, power supply (battery (B), AA and LR20 cells (AA-B or LR20-B), battery with solar panel (B/S), wired connection (W) and energy autonomy in parenthesis when concerned and available (in days or hours)), pre-processing (Z, A or C weighting (Z-w, A-w or C-w), gain amplification (G), calibration (Cal), frequency equalization (Eq), 1/3 or 1/1 octave band analysis (1/3 or 1/1), encoding (enc)), price (in Euros (EUR), United States dollars (USD), Great Britain pounds (GBP)), goal of the corresponding article (proof-of-concept (POC), sensor design only, operational WSN (Op-WSN)).

Reference	Node Plateform	MCU	N-2-S	Mic.	ADC	Power	Pre-Processing	Cost	Goal
Barham and Goldsmith [45] (2008)	FPGA		GSM	a-M		B (15d)	A-w, C-w, Tc	100 EUR	Op-WSN
Santini et al. [42] (2008)	Tmote	16-bits	802.15.4	ECM	12-bit (8 kHz)	AA-B			POC
McDonald et al. [12] (2008)	Triton	32-bits	802.11b	ECM	16-bit (49 kHz)		A-w	130 GBP	Op-WSN
Hakala et al. [49] (2010)	CiNet	8-bits	802.15.4	ECM	10-bit (33 kHz)	AA-B (ds)	A-w, G, Cal		Op-WSN
Tan and Jarvis [54] (2013)	TelosB	16-bits	802.15.4	ECM	12-bit (33 kHz)	B/S			POC
Tan and Jarvis [56] (2014)	TelosB	16-bits	802.15.4	a-M	12-bit (33 kHz)	B/S			POC
Segura-Garcia et al. [59] (2015)	Tmote	16-bits	802.15.4	ECM	12-bit (8/20 kHz)	B (78d)	Cal	41.45 EUR	POC
Segura-Garcia et al. [59] (2015)	R-Pi	32-bits	802.11	ECM	16-bit (22.05 kHz)	LR20-B (39h)	Cal		Op-WSN
Noriega-Linares and Navarro Ruiz [61] (2016)	R-Pi	32-bits	wired LAN	ECM		W	Cal, Eq, 1/3	121 USD	POC
Alsina-Pagès et al. [37] (2016)	NXP chip	32-bits	Wi-Fi/GSM	ECM	12-bit (nc)	B			Design only
Mydlarz et al. [62] (2017)	mini-PC	32-bits	Wi-Fi	a-M	16-bit (44.1 kHz)	W	Eq	100 USD	POC
Risojević et al. [38] (2018)	STM32F0 series	32-bits	ZigBee	a-M		B (7d)	A-w, G, Cal	41.45 EUR	Op-WSN
Peckens et al. [63] (2018)	Teensy USB	32-bits	XBee	ECM	16-bit (20 kHz)	B (7d)	A-w, G, Cal	135 USD	POC
Ardouin et al. [67] (2018)	STM32L4 series	32-bits	802.15.4	d-M	16-bit (32 kHz)	B/S	A-w, 1/3, enc		POC
Ardouin et al. [67] (2018)	R-Pi	32-bits	802.15.4	d-M	16-bit (32 kHz)	W	A-w, 1/3, enc		POC
Silvaggio et al. [66] (2019)	mini-PC		GSM	d-M		B/S,W	A-w, 1/3		Op-WSN
Silvaggio et al. [66] (2019)	MCU		GSM	ECM		B/S,W	A-w, 1/3		Op-WSN
Mydlarz et al. [62] (2019)	R-Pi	32-bits	Wi-Fi/POE	d-M	16-bit (48 kHz)	W	A-w, C-w, 1/3	80 EUR	Op-WSN
López et al. [68] (2020)	DSP Board	32-bits	radio (868 MHz)	ECM	24-bit (108 kHz)	B	Z-w, A-w, C-w, 1/3, 1/1		Op-WSN

**Table 2 sensors-20-02256-t002:** Mean features of low-cost noise sensors detailed in the literature: frequency range, sound level dynamic, sound level range, residual noise, output acoustic indicators (equivalent sound level L${}_{\mathrm{eq},T}$ or L${}_{\mathrm{Aeq},T}$, maximum (Max) and minimum (Min) levels, percentiles L${}_{N}$, 1/3 or 1/1 octave band spectrum, audio signal, psychoacoustic metrics, Peak detection). Data with the symbol * design measured sensor performances while the other ones are estimated.

Reference	F-Range	Dynamic	L-Range	Residual Noise	Outputs
Barham and Goldsmith [45] (2008)	20–20k Hz	70 dB	30–100 dB	25 dB	L${}_{\mathrm{eq},T}$, L${}_{N}$ (10 mn)
Santini et al. [42] (2008)					L${}_{1s}$
McDonald et al. [12] (2008)					L${}_{\mathrm{Aeq},T}$
Hakala et al. [49] (2010)	<16.5 kHz		30–90 dB		L${}_{\mathrm{eq},125\mathrm{ms}}$, L${}_{\mathrm{eq},1s}$
Tan and Jarvis [54] (2013)	<5 kHz *		93 dB *	60 dB *	
Tan and Jarvis [56] (2014)			100 dB *	50–60 dB *	Peak
Segura-Garcia et al. [59] (2015)	<20 kHz	96 dB			Psychoacoustic metrics
Noriega-Linares and Navarro Ruiz [61] (2016)	125–8k Hz * (1/3)				L${}_{\mathrm{eq},T}$, L${}_{N}$ (*N*=10,50,90), 1/3
Alsina-Pagès et al. [37] (2016)					
Sevillano et al. [72] (2016)			35–115 dB		L${}_{\mathrm{eq},1s}$, audio
Piper et al. [48] (2017)					L${}_{\mathrm{Aeq},0.2s}$
Mydlarz et al. [62] (2019, 2017)	20–20k Hz	88.1 dBA		29.9 dBA *	Audio (10 s)
Risojević et al. [38] (2018)	up to 16 kHz *	72 dB	50–100 dB *		L${}_{\mathrm{eq},250\mathrm{ms}}$
Peckens et al. [63] (2018)	<10 kHz *	50 dB *		50 dB *	L${}_{\mathrm{eq},125\mathrm{ms}}$ (10 mn each 1 hour)
Ardouin et al. [67] (2018)	20–16k Hz		35–105 dBA		L${}_{\mathrm{eq},125\mathrm{ms}}$, L${}_{\mathrm{eq},1s}$, 1/3
Silvaggio et al. [66] (2019)	20–20k Hz	70 dB	30(40)–100(110) dB	30–35 dBA	L${}_{\mathrm{eq},1s}$, 1/3
Mydlarz et al. [62] (2019)			32–100 dBA		L${}_{\mathrm{eq},125\mathrm{ms}}$, L${}_{\mathrm{eq},1s}$, 1/3, audio (10s)
López et al. [68] (2020)	up to 8 kHz		39.1–120.1 dB		L${}_{\mathrm{eq},125\mathrm{ms}}$, L${}_{\mathrm{eq},1s}$, Peak, Max, Min, L${}_{N}$ (*N*=1,5,10,50,90,95,99), 1/3, 1/1

**Table 3 sensors-20-02256-t003:** Minimal and optimal expected characteristics for the noise sensors.

Property	Minimal Target	Optimal Target
Measurement range	30–105 dB(A)	30–105 dB(A)
Frequency range	100–12k Hz	100–16k Hz
Integrated sound level	L${}_{A,\mathrm{eq},1s}$	L${}_{A,\mathrm{eq},125\mathrm{ms}}$
		L${}_{A,\mathrm{eq},1s}$
Spectrum	None	1/3 octave bands
Measurement frequency		Continuous
Pre-processing	A-weighting	(A, Z)-weighting
	Calibration	Calibration
		1/3 octave bands analysis
		Frequency equalization
Other indicators		Source recognition
		Noise event detection
Additional sensors	Temperature	Temperature
	Hygrometry	Hygrometry
Price	50 EUR	150 EUR

**Table 4 sensors-20-02256-t004:** Radio transmission protocols specifications.

Protocol	Bluetooth [85]	Bluetooth LE [85]	Wi-Fi [86]	Wi-Fi [86]	Zigbee and 6LoWPAN [87]	LoRaWAN [88]	Sigfox [89]
Specification	802.15.1	802.15.1	802.11g	802.11n	802.15.4	LoRa Alliance	Sigfox
Frequency	2.4 GHz	2.4 GHz	2.4 GHz	2.4 GHz 5 GHz	868 MHz (EU) 915 MHz (US) 2.4 GHz	Sub-GHz ISM band 868 MHz in EU	Sub-GHz ISM band 868 MHz in EU
Range indoor (m)	30	10	25	50	30	>100	>100
Range max (m)	100	50	75	125	1500	>10,000	>10,000
Data speed max	3 Mbit/s	1 Mbit/s	54 Mbit/s	540 Mbit/s	250 kbit/s	11 kbit/s	100 bit/s
Data speed typ.	2.1 Mbit/s	270 kbit/s	25 Mbit/s	200 Mbit/s	150 kbit/s	300–11k bit/s	100 bit/s
Peak current	150 mA	20 mA	150 mA	150 mA	50 mA	25 mA	25 mA
Sleep current	5 mA	1 μA	100 μA	100 μA	5 μA	4 μA	4 μA
Battery life	Month	Year	Day	Day	Month/Year	Years	Years
Network topologies	Star	Star	Star		Star, Tree, Mesh	Star	Star
Applications	Headsets Computer peripherals	Mobile phones Sport trackers eHealth devices Wireless sensors	PC (networking) WLAN	Same as 802.11g with improved performances Outdoor LAN	Smart home Wireless sensor networks Smart metering	Smart building Smart city	Smart building Smart city

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
