# Peer review of "Low-Cost Sensors for Urban Noise Monitoring Networks—A Literature Review"

_sensors, 2020, doi:10.3390/s20082256_

Round 1

Reviewer 1 Report

This is very good survey but survey to the bone. I expected "specification of an optimal design" but text is not like that. Please change the title as the paper is good enough and doesn't need any fake boost. If I felt that its title is misleading then someone else might too feel disappoointed. It wouldn't be due to the paper quality, which is good, but due to missed expectations.

Paper is focused on noise measurements while title has broad claim of "environment monitoring". Please narrow the title.

At page 8 Authors also mention "802.15.4 modulation". 802.15.4 is IEEE standard and definitely not "modulation". There are many modulations described and referenced in that standard. Please clarify the phrase.

I've also found some typos:

  • R-Pi - full name is Raspberry Pi
  • 802.15.4 - might be confusing for non experts so I suggest to use developed name such as:  IEEE standard 802.15.4
  • Wifi - Wi-Fi
  • cheep - chip (I suppose)

Author Response

On the behalf of all my co-authors, and as corresponding author, it is my pleasure to transmit the revised version of our manuscript to the MDPI/SENSORS journal (note that the manuscript title has been changed according to a reviewer comment)

Manuscript ID: sensors-764940

Type of manuscript: Review

Title (old): Low-cost sensors for urban environment monitoring networks: specification of an optimal design from a review of the literature

Title (new title): Low-cost sensors for urban noise monitoring networks: a literature review 

Authors: Judicaël Picaut *, Arnaud Can, Nicolas Fortin, Jeremy Ardouin, Mathieu Lagrange

All the revisions in the manuscript and our responses to reviewer’s comments:

  • are detailed point-by-point in the attached file,
  • are highlighted in a PDF version of the revised manuscript.

Reviewer 2 Report

My comments to authors:

In the paper, the authors reviewed the literature about the monitoring networks for urban environment by using low-cost sensors. In addition, they described the problem of noise pollution assessment. Finally, they presented the challenges in massive deployment of low-cost noise sensors. In general, the paper is well written, but we think there are some problems. My major comments are given as below.

1) In the abstract, there are some mistakes, such as line 8.

2) In the section of 2.3, they presented some key aspects and design choices. However, there lacks the description about the framework from whole aspect.

3) In the section 3, the optimal noise sensors for low-cost networks were unclearly discussed. For example, the optimizing sensor locations and network deployment was presented while how to design a scheme to complete this deployment was not given clearly.

Author Response

(The authors gave the same response as above.)

Round 2

Reviewer 2 Report

I think that the manuscript has been significantly improved and now warrants publication in Sensors.